# Disparities between native americans and white individuals in global outcome trajectories over the 5 years after traumatic brain injury: A model systems study

**Jack D. Watson**[1,2], **Paul B. Perrin**[3,4*], **Juan Carlos Arango-Lasprilla**[5]

1 Informatics, Decision Enhancement, and Analytic Sciences Center, Salt Lake City Department of Veterans Affairs, Salt Lake City, Utah, United States of America, 2 Department of Internal Medicine, Division of Epidemiology, University of Utah School of, Medicine, Salt Lake City, Utah, United States of America, 3 School of Data Science and Department of Psychology, University of Virginia, Charlottesville, Virginia, United States of America, 4 Central Virginia Veterans Affairs Health Care System, 5 Department of Psychology, Virginia Commonwealth University, Richmond, Virginia, United States of America

* perrin@virginia.edu

**Data availability statement:** The data are publicly available through an external data request to the TBI Model System (TBIMS) National Data and Statistical Center (https://www.tbindsc.org/researchers.aspx).

**Funding:** This research was funded by the National Institute on Disability, Independent

## Abstract

### Purpose

Traumatic brain injury (TBI) can lead to a host of challenges and negatively impacts Native Americans more than any other ethnic group in the U.S. Despite this, little research exists on Native Americans with TBI. The current study examined disparities in global outcome trajectories (overall level of function post-injury) between Native Americans and White individuals in the 5 years following TBI and whether sociodemographic or injury-related characteristics could account for this disparity.

### Method

The current study used a sample of 75 Native Americans demographically matched by sex, age, and injury severity to 75 White individuals from the U.S. Traumatic Brian Injury Model Systems (TBIMS) database (n = 150). A series of hierarchical linear models (HLMs) was used to examine longitudinal global outcome trajectories between the two groups.

### Results

Native Americans showed lower global outcome scores than their White counterparts with this difference worsening (i.e., growing larger) over time. The difference in scores and the differential movement over time were associated with differences between the ethnic groups in employment at the time of injury, substance use patterns, and type of insurance.

### Conclusion

This study highlights the need for identification of and early intervention for risk factors that predict disparities in rehabilitation outcomes and points to the need for greater access to culturally informed care for Native Americans with TBI.

Living and Rehabilitation Research (NIDILRR) and by the U.S. Department of Veterans Affairs. The funders had no role in study design, data collection and analysis, decision to publish, or preparation of the manuscript.

**Competing interests:** The authors have declared that no competing interests exist.

## Introduction

Traumatic Brain Injury (TBI) is one of the leading causes of death and disability in the United States (U.S.) and worldwide across all age groups [1]. Within the U.S., TBI is also the leading cause of death and disability for children and young adults [1–3]. Centers for Disease Control and Prevention (CDC) data show more than 200,000 TBI-related hospitalizations occurred in 2020, with nearly 70,000 TBI-related deaths in 2021 [1]. The CDC also estimates that within the U.S., 2 million people each year die, visit the emergency room, or require hospitalization as a result of a TBI [1,4]. Age, particularly older age, and being male are both risk factors for experiencing a TBI [1,4], with men being twice as likely to be hospitalized and three times as likely to die following a TBI compared to women [1,4]. Falls, collisions with an object, and motor vehicle accidents are leading causes of TBI in the U.S., while suicide, unintentional falls, and motor vehicle accidents are the leading causes of TBI-related deaths in the U.S [1,4].

TBI places a significant burden on the healthcare system as it is often accompanied by a host of negative sequelae and comorbidities [3]. Many individuals see significant changes to their physical function and may experience impaired movement capability and balance, cognitive difficulties (e.g., memory problems, reduced processing speed), difficulty with fine motor tasks, and problems with toileting [1,4–7]. Injury severity is a key predictor for mortality and return to pre-injury social roles and function [6,8], and poorer physical functioning is associated with lower self-reported life satisfaction, greater severity and higher rates of negative mental health symptoms, and worse self-reported quality of life [9].

TBI often has negative consequences for the survivor's mental health [1,4]. Individuals with TBI frequently report worse life satisfaction, lower levels of community engagement and productivity, and less social interaction than the general population [10]. Experiencing a TBI also predisposes individuals to higher rates and greater severity of depression and anxiety symptoms [9,11,12]. As many as 50% of those who experience a moderate to severe TBI will be diagnosed with depression while 36% will receive an anxiety disorder diagnosis [11–13]. TBI is a leading cause of post-traumatic stress disorder (PTSD), with 24% of those experiencing a TBI also meeting criteria for PTSD at some point subsequent to the injury [14]. The potential for suicide is a major concern following TBI as suicidal ideation is present in roughly 25% of people with TBI, and individuals with TBI die by suicide at a rate of twice the general population [1,4,15].

The return to pre-injury roles, social interaction, function, and productivity can be quite difficult, particularly for individuals with more severe TBI and disability [10,16,17]. The Glasgow Outcome Scale-Extended (GOS-E; [18]) has been used extensively in the rehabilitation literature to assess global outcomes following traumatic injury, especially TBI [19–23]. Global outcome, or function, is a rating for overall level of disability and neurological function following TBI and is highly predictive of a host of important psychosocial and health metrics like depression, likelihood to return to work, community engagement, and satisfaction with life [18,24]. Global outcome is composed of seven domains: (1) consciousness, (2) independence at home, (3) independence outside home, (4) work, (5) social and leisure activities, (6) family and friendships, and (7) return to normal life. Combined, these seven domains provide a means of assessing an individual's ability to function, thus creating a general picture of the health and recovery of an individual following TBI [18,24]. Global outcome is one of the key metrics for assessing rehabilitation post-injury as it is a vital indicator of a plethora of other important rehabilitation constructs like independence, community engagement, and dependency [18–20,24] and often serves as the primary outcome measure for clinical rehabilitation and research following TBI [25].

## TBI in native Americans

TBI disproportionately affects Native Americans more than any other ethnic group in the U.S. [1,4,26]. Native Americans experience the highest incidence of TBI, highest mortality rates post-TBI, highest rates of hospitalization post-TBI, and least amount of healthcare service utilization post-rehabilitation of all ethnic minorities in the U.S. [1,4,26–29]. Despite this, research on Native Americans with TBI is nearly nonexistent. Two systematic reviews conducted in 2017 revealed only eight studies regarding TBI in Native Americans, several of which were epidemiological in nature or had very small sample sizes [30,31]. As such, almost no research exists on rehabilitation outcomes or health disparities for Native Americans with TBI [28,30–32]. As of 2024, only two studies have used a demographically matched sample of Native Americans and White individuals to examine outcome disparities following TBI [32,33].

No research to date has attempted to fully investigate causes of the high rates of TBI within the Native American population. The limited research indicates that higher rates of violence, especially intimate partner violence (IPV), may play a role as Native American women experience IPV at rates five times the national average [34,35] and experience violence at a rate higher than any other ethnic gender group in the U.S. [36]. Native Americans are 300% more likely to experience violence as a factor contributing to TBI than any other ethnic group in the U.S. [4,37]. Substance use is another uniquely significant contributing factor for TBI in Native Americans [37–40], with rates twice that of other ethnic groups [40]. As substance use may impair an individual, thereby leading to an increased risk for TBI (e.g., driving under the influence), it is possible that differential rates of substance use may partly account for the higher rates of TBI within Native Americans [41].

Only a few studies have investigated longitudinal outcomes following TBI for Native Americans [32,33,42]. This research shows that Native Americans with TBI are likely to develop depression and anxiety symptoms [43], experience barriers to healthcare service utilization and acquiring stable, long-term employment, and face reductions in independence with an increased need for supervision or long-term care/assistance [28,32,33,44]. Research also suggests that the presence of a TBI magnifies many of the disparities experienced by the Native American population (e.g., higher rates of poverty; [45]). Only two studies have compared Native Americans to a demographically matched sample of White individuals with TBI to examine longitudinal outcomes and health disparities [32,33]. These studies found that Native Americans experienced lower community participation (Out and About) and functional independence following TBI than White individuals, particularly cognitive function, which declined over time for Native Americans while White individuals saw a slight increase [32,33]. Both employment status at the time of injury and the type of health insurance (private vs other) held by the person with TBI were significant factors predicting cognitive function [32], and educational attainment and type of insurance were significant factors predicting the Out and About component for community participation [33].

In addition to inequal outcomes following TBI, Native Americans are recognized as a severely underserved and marginalized population, facing disparities in diseases, healthcare utilization, poverty rates, and lower education and employment [45]. Native Americans are significantly more likely to face premature mortality due in large part to disparities in chronic disease and lack of appropriate specialized healthcare providers in tribal lands or through the Indian Health Service [46]. The lack of reliable and adequate healthcare services, heightened rates of high mortality-related injuries and diseases (e.g., TBI, diabetes), and suspicion of western medical practices and provider distrust, combine to create a bleak picture for the state of Native American healthcare [47,48].

The profound lack of research on the experience of Native Americans with TBI has resulted in large gaps in the knowledge of what challenges Native Americans face following TBI, specific areas of strength to bolster during rehabilitation, and unique cultural and contextual factors that may impact rehabilitation trajectories [30,31,39,40,43]. Further, such gaps may impact the ability of rehabilitation clinicians to provide culturally sensitive and specific care [49,50]. One such gap in the literature is longitudinal global outcome trajectories which, at present, have not been investigated for Native Americans with TBI, nor has any research examined possible disparities between Native American and White people in global outcome trajectories following TBI.

## Study purpose

The literature for differential outcomes during and following rehabilitation is growing but still relatively small with very few theoretical models conceptualizing the possible structures and origin of ethnic health disparities for individuals with TBI [40,49]. To date, the only theoretical model for healthcare disparities that was specifically constructed for use within the disability community is the Model of Healthcare Disparities and Disability (MHDD; [49]). In this model, Meade and colleagues argue that functional outcomes result from the intersection of (a) disability characteristics (e.g., clinical severity), (b) sociodemographic factors (e.g., employment status), and (c) intrinsic personal characteristics (e.g., sex). Thus, only with the consideration of these important intersectional characteristics can functional outcome following injury and disability be more fully understood [40,49,50]. Given the extensive differences in the causes of TBI, comorbidities, and outcomes between Native Americans and White individuals, the current study utilized a modified version of the MHDD which has been previously used to examine disparities between Native Americans and White individuals with TBI [32] to investigate the degree to which disparities in global outcomes exist between the two groups over the 5 years post-injury. Further, only one other study to date has investigated how demographic and TBI injury characteristics differ between Native Americans and White individuals in a demographically matched sample [32]. The current study highlighted demographic and injury-related characteristics that differed between the two groups then used these statistically significant differences to attempt to account for the disparity in global outcomes.

## Study rationale and hypotheses

While no research to date has investigated global outcome for Native Americans, previously literature has shown that Native Americans face a plethora of barriers to recovery including: higher rates of depression and anxiety, lower healthcare utilization and access, and lower functional independence following TBI [28,32,43,44], indicating Native Americans are at a greater risk for negative complications following TBI when compared to White individuals. Further, while little research to date has investigated possible predictors of outcomes following TBI for Native Americans, the literature for outcomes following TBI for other underserved populations (e.g., Hispanics) is growing, with evidence for the important of sociodemographic factors like employment, alcohol use, marital status, and violence as a cause of injury [51–54]. Based on this literature, we hypothesize that: (1) Native Americans with TBI will show worse global outcome scores than their demographically matched White counterparts. (2) The difference in global outcome scores will grow larger over time. And, finally, (3) even after accounting for the sociodemographic and injury-related covariates that differ significantly between the two groups, the disparity in global outcomes will still exist due to culturally and contextually specific variables not measured in the current study (e.g., multiple comorbidities, geographic distance to the nearest healthcare center, systemic racism).

## Method

### Procedure and study approval

The current study used data from the Traumatic Brian Injury Model Systems (TBIMS) U.S. National Database. Each of the 16 rehabilitation centers associated with the TBIMS have their own institutional review board (IRB) overseeing the study at their site, and therefore all data collection was conducted with IRB approval. The TBIMS is the largest longitudinal study of TBI outcomes in the world and collects data on a variety of factors and constructs including pre-injury information, sociodemographic characteristics, injury-related variables, physical and mental health information and diagnoses, and rehabilitation outcomes. At present, no other database offers such nuanced data on Native Americans with TBI. Data collection starts during inpatient rehabilitation after participants or a legal proxy complete informed consent, and follow-up data collections are attempted at 1, 2, 5, and every 5 years thereafter following discharge. Data can be drawn from direct patient interviews or medical examinations, medical record review, or data collections forms and can occur in person or via telephone with the participant or a person intimately familiar with the participant (e.g., informal caregiver). For the current study, only data for the first five years following rehabilitation were used due to the small sample of Native Americans with data 10 years and beyond. The U.S. Department of Health and Human Services funds the TBIMS program through the National Institute on Disability, Independent Living, and Rehabilitation Research (NIDILRR; [55]). The data are publicly available through an external data request to the TBIMS National Data and Statistical Center.

### Participants

Individuals were eligible for enrollment in the TBIMS only if they sustained a complicated mild, moderate, or severe TBI. They also had to be 16 years of age or older at injury, receive all subsequent inpatient rehabilitation at a TBIMS rehabilitation center, and be admitted for care within 72 hours of sustaining the injury. The injury had to result in: (1) Glasgow Coma Score (GCS) of less than 13 at the time of admission, (2) unconsciousness for longer than 30 minutes, (3) posttraumatic amnesia (PTA) for longer than 24 hours, or (4) trauma-related intracranial abnormality on neuroimaging.

The current study included 75 Native Americans and 75 White individuals with TBI for a total sample of 150 (Table 1). There were initially 90 Native Americans in the TBIMS database; however, only 75 had at least one complete data point (year 1, 2, or 5) for global outcome. These 75 Native Americans were then demographically matched by the researcher to 75 White individuals. All participants were matched by sex, injury severity (time spent in PTA), and age (+/- 1 year). The White person matching all three categories also needed at least one complete data point for global outcome. Matching priority was given to individuals whose data completeness most closely mirrored the Native American participant (i.e., if the Native American had only year 1 data for global outcome, then matching priority was given to a White person who also had only year 1 data). If multiple people fulfilled the matching criteria, the person who appeared first in the database was selected. Demographic and injury-related characteristics, separated by ethnicity, appear in Table 1 and information on data missingness can be found in Table 2.

### Study measures

**Demographic and injury-related variables.** The TBIMS database collects a vast array of demographic, injury-related, health, and outcome variables. The current study examined the following variables as possible covariates: age, sex, injury severity (defined

**Table 1.** Characteristics of Individuals with TBI.

| Variable | Native American | White | p-value |
|---|---|---|---|
| Age at Injury, M (*SD*) | 38.52 (15.61) | 38.53 (15.57) | p = .996 |
| Sex, n (%) | | | p = 1.000 |
| Male | 51 (68.00) | 51 (68.00) | |
| Female | 24 (32.00) | 24 (32.00) | |
| Years of Education Pre-Injury, M (*SD*) | 11.13 (2.58) | 12.05 (3.04) | p = .085 |
| Employment at Injury, n (%) | | | p = .006 |
| Employed | 33 (44.00) | 50 (66.70) | |
| Not Employed | 35 (46.70) | 20 (26.70) | |
| Annual Earning, n (%) | | | p = .626 |
| <9,999 | 7 (9.30) | 10 (13.30) | |
| 10,000-19,999 | 9 (12.00) | 6 (8.00) | |
| 20,000-29,999 | 9 (12.00) | 12 (16.00) | |
| 30,000-39,999 | 4 (5.30) | 2 (2.70) | |
| 40,000-49,999 | 3 (4.00) | 3 4.00) | |
| 50,000-59,999 | – | 3 (4.00) | |
| 60,000-69,999 | 1 (1.30) | 2 (2.70) | |
| 70,000-79,999 | 2 (2.70) | 2 (2.70) | |
| 80,00- 89,999 | 1 (1.30) | – | |
| 90,000-99,999 | 1 (1.30) | 1 (1.30) | |
| > 100,000 | 1 (1.30) | 5 (6.70) | |
| Type of Work, n (%) | | | p = .541 |
| Blue Collar | 29 (38.70) | 39 (52.00) | |
| White Collar | 7 (9.30) | 13 (17.30) | |
| Cause of Injury, n (%) | | | p = .092 |
| Non-Violent | 65 (86.70) | 71 (94.70) | |
| Violent | 10 (13.30) | 4 (5.30) | |
| Insurance Type, n (%) | | | p = .032 |
| Private | 28 (37.80) | 41 (55.40) | |
| Non-Private | 46 (62.20) | 33 (44.60) | |
| Marital Status, n (%) | | | p = .299 |
| Married | 28 (37.30) | 22 (29.30) | |
| Not Married | 47 (62.70) | 53 (70.70) | |
| Language Spoken at Home, n (%) | | | p = .149 |
| English | 56 (74.70) | 63 (84.00) | |
| Other than English | 4 (5.30) | 1 (1.30) | |
| Illicit/Non-Prescription Drug Use, n (%) | | | p = .014 |
| Reported Problematic Use | 45 (64.30) | 31 (43.70) | |
| Did Not Report Problematic Use | 25 (35.70) | 40 (56.30) | |
| Cigarette Use, n (%) | | | p = .558 |
| Smoked Prior to Injury | 9 (12.00) | 1 (1.30) | |
| Did Not Smoke Prior to Injury | 12 (16.00) | 4 (5.30) | |
| Alcohol Use, n (%) | | | p = .025 |
| Reported Problematic Use | 41 (54.70) | 20 (26.70) | |
| Did Not Report Problematic Use | 35 (46.70) | 53 (70.70) | |
| Days Spent in PTA, M (*SD*) | 23.31 (25.42) | 23.80 (22.75) | p = .901 |

**Note.** Not all categories have *n* = 75 due to missing or refused data. Mean and standard deviation are included for continuous variables and *n* and percentage for categorical variables.

as time spent in PTA), employment at the time of injury (employed vs not employed), type of employment (blue vs white collar), marital status (married vs not married), type of insurance (private vs other), language spoken at home (English vs other), cause of injury (violent vs non-violent), country of origin (U.S. vs other), annual earnings, education, substance use (reported using illicit/non-prescription drug use in the month before injury vs denied use), cigarette use (reported smoking in the month prior to injury vs denied smoking), and alcohol use (reported binge drinking in the month before injury vs denied binge drinking). For data collection within the TBIMS, binge drinking is defined as consuming 5 or more alcoholic drinks in one sitting for a male and 4 or more drinks in one sitting for a female. For analyses, all variables were either dichotomized (e.g., cause of injury, marital status) or given a reference point of zero if left as a continuous variable (e.g., annual earnings).

These constructs were chosen as they have a long history of use within the rehabilitation literature as key predictors of health outcomes and important considerations for health disparities following traumatic injury [18,25,49,51,56,57]. These variables also mirrored the theoretical model of the MHDD (Fig 1) and were consistently collected by the TBIMS database since its inception, thereby allowing retention of as many participants as possible while still making full use of the MHDD.

**Global outcome. Glasgow Outcome Scale-Extended (GOS-E; 18).** The GOS-E is a structured interview of either the individual with TBI or a reliable informant (e.g., caregiver) that assess individuals across the 7 domains of global function. The global outcome score is based on the overall score for the lowest of the seven categories. Scores are: 1 = dead; 2 = vegetative state; 3 = lower severe disability; 4 = upper severe disability; 5 = lower moderate disability; 6 = upper moderate disability; 7 = lower good recovery; 8 = upper good recovery [18]. As such, GOS-E total scores were used for the current study. The GOS-E is a widely accepted measure within rehabilitation research and has demonstrated adequate validity and reliability [18–23,58].

## Data analysis

**Preliminary and curvature analyses.** Multicollinearity was assessed for all covariates with a critical cut off of $r > 0.70$. Means, standard deviations, and frequencies were examined for all sociodemographic and injury-related characteristics and appear in Table 1 separated by ethnicity. ANOVA or $\chi^2$ analyses were used to determine which variables differed significantly between the two races/ethnicities and also appear in Table 1. Normality tests on global outcome were assessed with a critical cutoff for skewness and kurtosis set at > 1.5. The percent of missing data at years 1, 2, and 5 post-discharge was also calculated (Table 2), and Little's Missing Completely at Random (MCAR) was used to assess whether data was missing completely at random. As long as the participant had at least one data point for global outcome, full information maximum likelihood (FIML) estimation was used to retain all participants ($n = 150$) despite missing data.

An unconditional growth model was conducted via hierarchical linear modeling (HLM) to determine the best curvature model for the study analyses. Two HLMs were run. The first used the intercept and time as fixed effect predictors and the second added time*time to determine whether a linear (straight line) or quadratic (u-shaped) model best fit the data. A -2 log likelihood (-2LL) for each successive model with a critical $\chi^2$ value of significant difference at $\alpha = .05$ and $> 3.84$ drop from the previous model (at 1 degree of freedom) was used to determine the best fitting model curvature.

**Table 2. Global Outcome by Race.**

| Variable | Native American | | | White | | |
|---|---|---|---|---|---|---|
| | *M (SD)* | # With Data | % Missing | *M (SD)* | # With Data | % Missing |
| One-Year GOSE | 5.42 (1.84) | 69 | 8.00 | 5.72 (1.68) | 68 | 9.30 |
| Two-Year GOSE | 5.42 (1.80) | 67 | 10.70 | 5.78 (1.77) | 68 | 9.30 |
| Five-Year GOSE | 4.95 (1.98) | 44 | 41.30 | 5.98 (1.65) | 63 | 16.00 |

**Primary analyses. Primary Set 1** The first set of HLMs assessed whether there was a difference in global outcome trajectories over the 5 years after injury between Native Americans and White individuals with TBI. A follow-up HLM was then conducted utilizing an interaction term between race/ethnicity and time (ethnicity*time) to see if the difference in global outcome trajectories worsened over time.

**Primary Set 2** The second set of HLMs repeated the analyses from the first set but incorporated the significantly different sociodemographic and injury-related covariates to determine if they accounted for the difference in global outcome scores between the two groups.

## Results

### Normality, sociodemographic differences, and curvature analyses

No problematic multicollinearity was found between the sociodemographic and injury-related variables (all $r < 0.70$). ANOVAs and $\chi^2$ tests indicated that Native Americans were less likely to be employed at the time of injury than White individuals, less likely to have private insurance, more likely to report having used illicit/non-prescription drugs in the month prior to injury, and more likely to report binge drinking. All other examined sociodemographic and injury-related variables were not significantly different between the two groups (Table 1).

Means and standard deviations for global outcome, separated by race/ethnicity, are reported in Table 2. Normality tests for global outcome revealed acceptable skewness and kurtosis (all values $< 1.25$), and examination of the -2LL for the curvature analysis revealed that a linear model best fit the data. Little's MCAR test indicated that the data were missing completely at random ($\chi^2$ [9] = 8.03, $p$ = .532), and HLM's FIML procedure was utilized to retain all participants and avoid possible sampling bias that occurs through listwise deletion or attrition. Thus, all 150 participants were retained despite data missingness (Table 2). The use of FIML to retain participants is a common practice in studies utilizing HLM, particularly within the rehabilitation literature [32,51,59–61].

### Primary set 1

The first set of HLMs (Table 3) revealed there was a significant main effect of ethnicity on global outcome trajectories ($p$ = 0.041), indicating Native Americans generally had lower global outcomes scores compared to White individuals (Fig 2; hypothesis 1).

There was also a significant interaction effect between race/ethnicity and time (Table 3), indicating global outcomes scores changed over time differentially as a function of ethnicity ($p$ = 0.009), with Native Americans worsening over time while White individuals improved (Fig 2; Table 2; hypothesis 2).

**Effect Sizes and Post-hoc Power Analysis** Cohen's *d* effect sizes were calculated for the differences between Native American and White Individuals' GOS-E scores at years 1, 2, and 5 post-injury (Table 4). Year one was below the threshold of a small-sized effect; year two showed a small-sized effect, and year five showed a medium sized-effect. A post-hoc power analysis

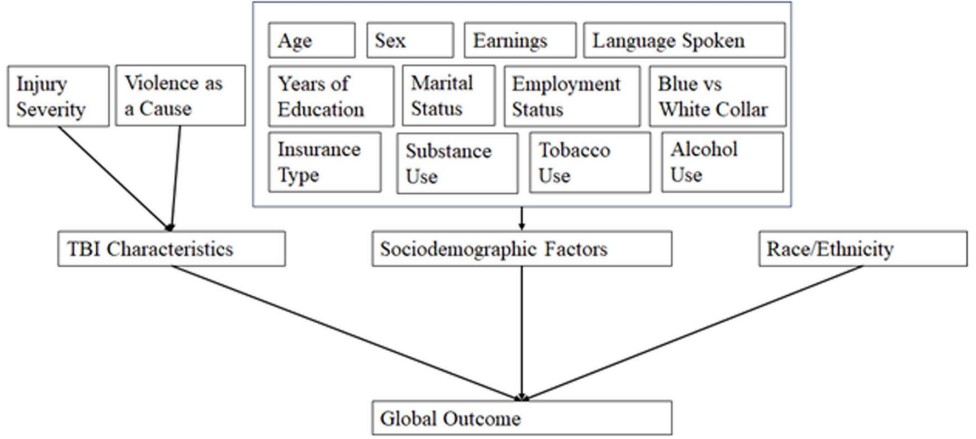

**Fig 1. Current study theoretical model.** This model is an adaptation of the MHDD.

conducted in G*Power 3.1.9.7 targeting a medium effect size with α = .05 and power at 0.80, revealed the need for a total sample size at $N = 102$, well below the 150 sample for the current study.

## Primary set 2

For the second set of HLMs, the significantly different sociodemographic covariates (employment at the time of injury, insurance type, illicit/non-prescription drug use within the month before injury, and binge drinking within the month before injury) were entered into the models to determine if these variables accounted for the difference in global outcome trajectories over time (Table 3; hypothesis 3). Results indicated that these covariates accounted for the difference in global outcome scores between the two groups, decreasing the significance of ethnicity as a predictor ($p$ = 0.348; Table 3; hypothesis 3); thus, for the study's third hypothesis, we failed to reject the null. Type of insurance remained the only significant predictor of global outcome scores ($p < 0.001$; Table 3), suggesting it was a primary driver of this racial/ethnic outcome disparity.

Because there was a significant interaction effect between ethnicity and time (i.e., the trajectory slopes differed; Fig 2), another follow-up HLM was run to see if the covariates that differed significantly between the two groups could also account for the differential change in global outcome trajectories. Again, with the inclusion of the covariates, the time*ethnicity interaction term decreased in significance ($p$ = 0.139), suggesting the covariates accounted for the significant differential change in global outcomes scores over time as a function of race/ethnicity. Again, type of insurance remained the only significant predictor and was a primary driver of the disparity ($p < 0.001$).

## Discussion

The current study's primary aim was to determine whether a disparity in global outcome trajectories existed between Native Americans and White individuals with TBI over the 5 years after injury and to determine if sociodemographic and injury-related characteristics that differed significantly between the two groups could account for the disparity. ANOVA and $\chi^2$ tests indicated that Native Americans were less likely to be employed at the time of injury than White individuals, less likely to have private insurance, more likely to report having used

**Table 3. Predictors of Global Outcome via HLM.**

| Predictor | GOS-E | |
|---|---|---|
| | *b*-weight | *p*-value |
| Set 1: Race | | |
| Intercept | 5.83 | <.001 |
| Time | -0.03 | 0.470 |
| Native American vs. White | -0.51 | 0.041* |
| Set 1: Race Interaction with Time | | |
| Intercept | 5.69 | < 0.001 |
| Time | 0.06 | 0.248 |
| Native American vs. White | -0.21 | 0.443 |
| Time*Race | -0.21 | 0.009* |
| Set 2: Race with Covariates | | |
| Intercept | 5.02 | < 0.001 |
| Time | -0.02 | 0.650 |
| Native American vs. White | -0.26 | 0.348 |
| Employment at Injury | 0.14 | 0.626 |
| Type of Insurance | 0.93 | < 0.001** |
| Drug Use | 0.13 | 0.661 |
| Alcohol Use | -0.02 | 0.195 |
| Set 2: Race with Covariates and Time Interactions | | |
| Intercept | 4.94 | < 0.001 |
| Time | 0.03 | 0.555 |
| Native American vs. White | -0.07 | 0.803 |
| Employment at Injury | 0.14 | 0.619 |
| Type of Insurance | 0.93 | < 0.001** |
| Drug Use | 0.12 | 0.670 |
| Alcohol Use | -0.02 | 0.203 |
| Time*Race | -0.13 | 0.139 |

*Indicates significance at 0.05.

**Indicates significance at 0.001

illicit/non-prescription drugs in the month prior to injury, and more likely to report binge drinking in the month prior to injury. Native Americans had generally worse global outcome scores over the five years following TBI, and the gap between Native Americans and White individuals grew larger (i.e., worsened) linearly over time (Fig 2). The difference in global outcome scores between the two groups, both generally and differentially over time, dissipated after the addition of the sociodemographic characteristics that differed significantly between the two groups.

## Sociodemographic differences

In line with previous research suggesting Native Americans experience lower employment rates than the general U.S. population and within the TBI population [32,33,45], the current study found that Native Americans were significantly less likely to be employed prior to experiencing a TBI than their White counterparts. While information on employment and insurance rates for Native Americans is lacking, particularly for Native Americans

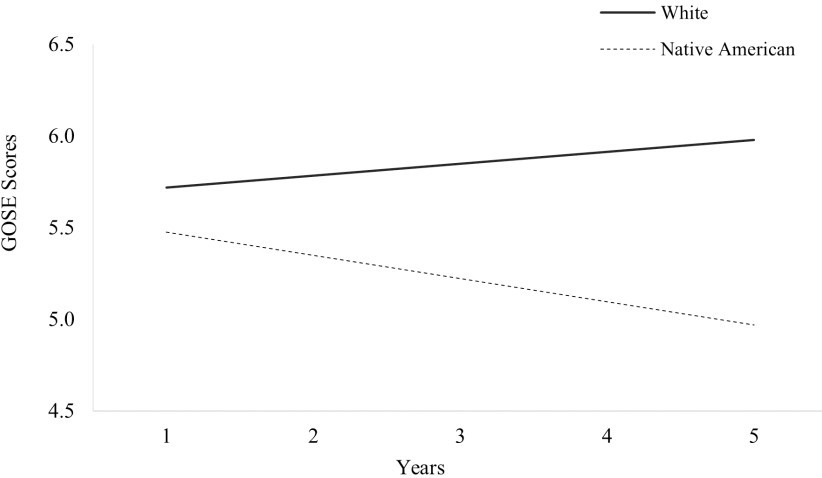

**Fig 2. Main effect of race/ethnicity on global outcome trajectories over the 5 years after injury.** $n$ = 150, although means were calculated from available data at each time point.

**Table 4. Effect Sizes for GOSE at 1-, 2-, and 5-Years Post-injury by Race/Ethnicity.**

| Variable | | Native Americans | White Individuals |
|---|---|---|---|
| | Cohen's $d$ | M (SD) | M (SD) |
| One Year GOSE | 0.17 | 5.42 (1.84) | 5.72 (1.68) |
| Two Year GOSE | 0.20 | 5.42 (1.80) | 5.78 (1.77) |
| Five Year GOSE | 0.58 | 4.95 (1.98) | 5.98 (1.65) |

with TBI, a separate TBIMS study utilizing a sample that partially overlaps with the present study also suggested Native Americans were less likely to be employed prior to injury and less likely to have private insurance than White individuals [32]. These may be key contributing factors for why Native Americans are more likely to experience barriers to healthcare access and less likely to engage with the healthcare system as private insurance often affords greater and more varied access to care, and employment is often a primary means of acquiring health insurance [28,62]. Private health insurance also typically affords greater and more varied access to healthcare than non-private alternatives. It is possible that Native Americans who lack private healthcare experience worse care or have a more difficult time accessing appropriate care, thus negatively impacting their recovery post-TBI. Given the higher rates of poverty within the Native American population coupled with their documented concerns about the high cost of healthcare, this may make Native Americans less likely to seek healthcare services even if they are available [48,63]. Issues such as cultural dissonance, provider distrust, and suspicion of Western medical practices may further impede Native Americans' decision to seek treatment [47,48]. These concerns, coupled with higher rates of TBI and chronic comorbid conditions (e.g., diabetes) create a scenario in which Native Americans may need greater and more specialized healthcare but be less likely to engage [32,33,47,63].

The research on substance use rates within the Native American population is mixed with some studies suggesting Native Americans have higher rates of substance use than the general population [45] and more recent research indicating comparable rates [64]. However, the rehabilitation literature indicates Native Americans are far more likely to have substance

use as a contributing factor to their TBI than all other racial/ethnic groups [37–40]. Thus, the current study's findings that Native Americans were significantly more likely to report illicit/non-prescription drug use or binge drinking in the month prior to experiencing a TBI is in line with the general rehabilitation literature. It is important to note, however, that the current study did not examine if substance use was a causal factor of TBI within this sample. Substance use (both illicit non-prescription drugs and alcohol use) in the month prior to experiencing a TBI was only examined as a possible predictor of disparities in global outcome.

## Global outcome

Results suggested that Native Americans with TBI had lower global outcome scores compared to White individuals with TBI and that this difference worsens over time (Fig 2). This is an important rehabilitation disparity as global outcome is a key metric of rehabilitation success as well as a primary indicator of likelihood to return to previous roles and function [18–20,24,25]. Both the difference in global outcome scores between Native Americans and White individuals and the change over time dissipated after the addition of the sociodemographic characteristics that differed significantly between the two groups (employment at the time of injury, insurance type, illicit/non-prescription drug use within the month before injury, and binge drinking within the month before injury). Type of insurance (private vs other; $p < .001$) appears to be the primary driver of disparities in global outcome trajectories for the models including the significantly different covariates (Table 3).

Gainful employment can provide a number of important benefits including income, a sense of belonging and purpose, and a source of social interaction [65]. This is especially important as unemployment has been linked to a plethora of negative mental health constructs (e.g., depression; [65]). Further, rehabilitation literature on employment prior to TBI indicates it is an important metric for predicting both motor and cognitive function post-injury [32,51]. It is, therefore, not surprising that employment status prior to injury may partly account for the lower global outcome scores for Native Americans with TBI compared to their White counterparts.

Substance use, both alcohol and illicit/non-prescription drugs, has been shown to be a uniquely significant contributing factor to TBI for Native Americans [37–40]. Previous research utilizing the TBIMS database on outcome disparities between Native Americans and White individuals following TBI indicated that alcohol use could have been an important factor predicting differences in motor and cognitive function but suggested the sample size was too small to accurately determine this [32]. The current study builds on this literature by demonstrating that substance use in the month preceding experiencing a TBI may partly account for disparities in global outcomes between Native Americans and White individuals.

Type of insurance was also identified in the current study as a key source of disparities in global outcome scores between Native Americans and White individuals with TBI. This makes sense as private insurance can provide greater access to care and more varied resources. The lack of resources may result in less specialized care or force Native Americans to use rural healthcare centers or Indian Health Service facilities that may be ill-equipped to provide the type and quality of care necessary for more complex conditions like TBI [47]. Given that Native Americans frequently experience barriers to healthcare services (e.g., cultural mistrust, racism) and are less likely to utilize healthcare services and resources and to receive or be recommended for more advanced services post-rehabilitation [28,38], it is possible that type of insurance may limit access for Native Americans and thereby reduce global

outcome scores. While not assessed in the present study, it is possible the lack of private insurance and reduced healthcare service utilization may lead to the development of additional comorbid conditions [46].

Finally, while not assessed in the present study, a plethora of research has indicated racism as a key factor contributing to healthcare inequalities [66–68]. Especially within the U.S., some research has suggested that racism is embedded within institutions, legal systems, and social structures and continues to negatively impact racial/ethnic minority communities who are unfairly targeted or marginalized by the U.S. healthcare system [69]. The limited literature on Native Americans with TBI indicates they experience high levels of racial discrimination [70] and are more likely to receive invasive surgery whereas White individuals are less likely and instead to be recommended for less invasive treatments like occupational or physical therapy, despite comparable TBI pathology [38]. This, coupled with disparities in various outcomes (e.g., cognitive and motor function, depression) can paint a bleak picture for the state of care received by Native Americans and may be contributing to lower global outcome scores [32,43].

The MHDD theorizes that health-related outcomes for individuals with disability are the result of interactions between a wide variety of factors. The current study reinforces this model by providing a picture of disparities in global outcome between Native Americans and White individuals with TBI that is complex and intersectional. Employment probabilities, type of insurance, illicit/non-prescription drug use, problematic alcohol use, and systemic barriers may all play a role in predicting global outcome scores and disparities for Native Americans with TBI. Type of insurance appears to be particularly important as it was the only predictor to remain significant across all models.

## Clinical implications

Native Americans are one of the most underserved populations in the U.S., and research on TBI for Native Americans is nearly non-existent [30,43,45]. This makes it challenging to accurately care for Native Americans and nearly impossible to provide culturally sensitive rehabilitation programs. Research with racial/ethnic minority populations with disability has called for a specific investigation of the unique cultural and contextual factors underpinning rehabilitation with a particular emphasis on race/ethnicity [49,50], and this study answers that call by being the first to investigate longitudinal trajectories in global outcomes following TBI for Native Americans. Given that global outcomes are such an important metric for predicting rehabilitation success and return to pre-injury roles and function [18–20,24,25], the current study highlights an important outcome disparity between Native Americans and White individuals. Greater support, early intervention, and care that is culturally tailored to Native Americans may assist in closing the gap in global outcome scores.

The current study highlighted both unemployment and substance use (both alcohol and illicit/non-prescription drugs) in the month preceding a TBI as potentially accounting for some of the effect of race/ethnicity on global outcome trajectories for Native Americans compared to White individuals. These characteristics can be used by clinicians to identify Native Americans who may need greater support or intervention. Examples of possible interventions include employment assistance/training, substance use cessation programs or counseling, support groups, and financial assistance (e.g., Temporary Assistance for Needy Families).

Type of insurance was also predictive of global outcome trajectories. As noted earlier, the limited research on Native Americans with TBI indicates a stark difference in the ways in which Native Americans interact with and are treated by the U.S. healthcare system when

compared to White individuals [38,70]. Native Americans can experience significant cultural differences and provider distrust and may have suspicion of western medicine practices [48]. Their families are excluded from medical decision making and meetings at a rate that far exceeds White families [38]. Thus, it is possible that type of insurance may be indicative of both the amount of healthcare resources available to Native Americans as well as its quality (both its effectiveness and whether or not the treatment is culturally informed).

Thus, rehabilitation clinicians should make a concerted effort to assess patient health literacy and ensure a mutual understanding of healthcare goals. Including family members and other vital supporters in healthcare conversations and decision making could also be beneficial. This can help reinforce trust in the care that is provided as well as ensure that both the patient and their family feel heard, valued, and understood. Finally, clinicians and allied health professionals should conduct a detailed assessment of both the patient's needs as well as barriers to healthcare access and provide resources and advocacy where appropriate. This is especially important as the rehabilitation literature indicates Native Americans with TBI experience significant barriers to healthcare access and at least one other study from the TBIMS database using a similar sample found that type of insurance was a key predictor of cognitive and motor function [32].

## Limitations and future directions

When interpreting the findings of the current study, several limitations and future directions should be considered. While the current study identified employment at the time of injury, substance use, and type of insurance as important factors accounting for global outcome disparities between Native Americans and White individuals, the exact mechanism for these relations was not uncovered in the current study. Despite the large number of sociodemographic and injury-related characteristics investigated, it is impossible to prove a causal relationship between these covariates and global outcome disparities. That is, the significantly different variables between the two groups could be markers for other unmeasured sources of variance that directly contribute to disparities in global outcome following TBI for Native Americans. Future research may wish to examine causal pathways or, with a larger sample, account for additional covariates to gain a more accurate picture of the origin of outcome disparities for Native Americans with TBI. Such research would also help inform rehabilitation services and culturally sensitive care. Research indicates that other variables not assessed in the current study like systemic barriers (e.g., distance to the nearest hospital), quality of care provided, the number and severity of comorbid conditions, and/or cultural dissonance between patient and provider may be important factors contributing to differential outcomes following TBI [32,33,45,47,48]. Particularly given the limited research on racial experiences of Native Americans with healthcare systems and providers [70], future research may wish to investigate how racial bias may impact healthcare access and quality for Native Americans with TBI.

The present study examined fourteen sociodemographic and injury-related characteristics as possible covariates that could partially account for the inequality in global outcome; however, data for both annual earnings and tobacco use were limited. It is possible that a larger, more complete dataset could uncover more nuanced findings related to both of these variables. Tobacco use may be an especially important factor for understanding disparities following TBI as it can lead to additional comorbid conditions (e.g., emphysema, lung cancer) that might further hamper recovery. Differential rates of tobacco use might result in differential rates of comorbid conditions, a key predictor of mortality following TBI [8]. Further, the present study did not include a host of other possibly important variables (e.g., comorbid conditions, geographic location, or distance to the nearest healthcare or therapy

center) which may warrant investigation in future research. The use of the total score for the GOS-E reduces some of the nuance that may be obtained by exploring item-level differences. Future studies may wish to use a larger sample to analyze differences in specific components of the GOS-E.

While the sample size of 150 Native American and White individuals is adequate for the present analyses, large enough to detect a medium effect size, [71,72] and unique within the rehabilitation literature to date, it makes generalizing to larger populations of Native Americans somewhat difficult given the culturally distinct and geographically isolated nature of Native American tribes within the U.S. [45]. Further, the sample size is well below the sample size recommended by G*Power to detect a small sized effect ($N = 620$), further limiting the generalizability of the study's results. The use of FIML allowed the retention of individuals who were missing some data for global outcomes, particularly Native Americans at year 5. Thus, some of the study's findings may be due to differential attrition even though Little's MCAR suggested the data were missing completely at random. Future studies should include a larger sample of Native Americans with less missing data and identify participants by Native American tribe (an identifier not available in the TBIMS database) as such methods would help increase the generalizability of the results beyond those of the present study.

## Conclusion

This was the first study to examine disparities in longitudinal global outcome trajectories for Native Americans with TBI. Results showed that Native Americans had worse global outcome scores than White individuals and that this gap in scores worsened over the five years following TBI. Further, this global outcome disparity was accounted for by significant differences between the Native American and White groups in employment at the time of injury, illicit/non-prescription drug use in the month preceding TBI, binge drinking in the month preceding TBI, and type of insurance (private vs other). Type of insurance appeared to be particularly important as it was the only remaining significant predictor of global outcome trajectories across all models. This study fills an important gap in the limited research on TBI in Native Americans and highlights the need for identification of and early intervention for risk factors that predict inequalities in rehabilitation outcomes. Finally, the current study also points to the need for greater access to culturally informed care for Native Americans with TBI.

## Acknowledgement

The Traumatic Brain Injury (TBI) Model Systems National Database is a multicenter study of the TBI Model Systems Centers Program, and is supported by NIDILRR a center within the Administration for Community Living (ACL), Department of Health and Human Services (HHS). However, these contents do not necessarily reflect the opinions or views of the TBI Model Systems Centers, NIDILRR, ACL or HHS.

## Author contributions

**Conceptualization:** Jack D. Watson, Paul B. Perrin, Juan Carlos Arango-Lasprilla.

**Formal analysis:** Jack D. Watson, Paul B. Perrin.

**Investigation:** Jack D. Watson.

**Methodology:** Jack D. Watson, Paul B. Perrin.

**Supervision:** Paul B. Perrin, Juan Carlos Arango-Lasprilla.

**Validation:** Jack D. Watson.

**Visualization:** Jack D. Watson.

**Writing – original draft:** Jack D. Watson.

**Writing – review & editing:** Paul B. Perrin, Juan Carlos Arango-Lasprilla.

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
