## [Decision Letter · Decision Letter 0]

5 Sep 2024

PONE-D-24-26489Disparities Between Native Americans and White Individuals in Global Outcome Trajectories over the 5 Years after Traumatic Brain Injury: A Model Systems StudyPLOS ONE

Dear Dr. Perrin,

Thank you for submitting your manuscript to PLOS ONE. After careful consideration, we feel that it has merit but does not fully meet PLOS ONE’s publication criteria as it currently stands. Therefore, we invite you to submit a revised version of the manuscript that addresses the points raised during the review process. Comments from the editorial office: Upon internal evaluation of the reviews provided, we kindly request you to disregard the reviewer report provided by Reviewer 2. No amendments are required in response to Reviewer 2’s comments. ’

We look forward to receiving your revised manuscript.

Kind regards,

Annesha Sil, Ph.D.

Associate Editor

PLOS ONE

Journal Requirements: When submitting your revision, we need you to address these additional requirements. 1. Please ensure that your manuscript meets PLOS ONE's style requirements, including those for file naming. The PLOS ONE style templates can be found at https://journals.plos.org/plosone/s/file?id=wjVg/PLOSOne_formatting_sample_main_body.pdf and https://journals.plos.org/plosone/s/file?id=ba62/PLOSOne_formatting_sample_title_authors_affiliations.pdf 2. We note that the grant information you provided in the ‘Funding Information’ and ‘Financial Disclosure’ sections do not match.  When you resubmit, please ensure that you provide the correct grant numbers for the awards you received for your study in the ‘Funding Information’ section. 3. Thank you for stating the following financial disclosure: "This research was funded by the National Institute on Disability, Independent Living and Rehabilitation Research (NIDILRR) and by the U.S. Department of Veterans Affairs." Please state what role the funders took in the study.  If the funders had no role, please state: ""The funders had no role in study design, data collection and analysis, decision to publish, or preparation of the manuscript."" If this statement is not correct you must amend it as needed. Please include this amended Role of Funder statement in your cover letter; we will change the online submission form on your behalf. 4. Please note that in order to use the direct billing option the corresponding author must be affiliated with the chosen institute. Please either amend your manuscript to change the affiliation or corresponding author, or email us at plosone@plos.org with a request to remove this option.

Reviewers' comments:

Reviewer's Responses to Questions

**Comments to the Author**

1. Is the manuscript technically sound, and do the data support the conclusions?

Reviewer #1: Yes

Reviewer #2: Yes

Reviewer #3: Yes

Reviewer #4: No

2. Has the statistical analysis been performed appropriately and rigorously? 

Reviewer #1: Yes

Reviewer #2: I Don't Know

Reviewer #3: Yes

Reviewer #4: No

3. Have the authors made all data underlying the findings in their manuscript fully available?

Reviewer #1: Yes

Reviewer #2: Yes

Reviewer #3: Yes

Reviewer #4: Yes

4. Is the manuscript presented in an intelligible fashion and written in standard English?

Reviewer #1: Yes

Reviewer #2: Yes

Reviewer #3: Yes

Reviewer #4: Yes

5. Review Comments to the Author

Reviewer #1: This paper was easy to read, used appropriate methods to assess the hypotheses, and addressed limitations as well as addressed future directions for research and clinical considerations for health care professionals. As such, this paper should assist clinicians and researchers in this area for future research and improvement in healthcare delivery for the relevant populations.

Reviewer #2: 1. Issue of Missing Data:

The article mentions the use of FIML to address some issues of missing data. However, it does not elaborate on the specifics of the missing data, such as which variables are missing, the proportion of missing data, and whether the missing data could potentially affect the outcomes. It is recommended that the authors provide additional details on the missing data and discuss its potential impact on the results.

2. Problem of Sample Representativeness:

Although the sample size meets the analysis requirements, the cultural diversity of Native American tribes and their scattered geographical distribution may limit the representativeness of the sample. It is suggested that the authors discuss the issue of sample representativeness and explain the generalizability of their research findings.

3. Insufficient Exploration of Causal Mechanisms:

While the article identifies the impact of unemployment, substance use, and insurance type on global outcome trajectories, it does not delve into the underlying causal mechanisms. The authors are advised to explore, based on existing literature, how these factors might influence the recovery process of Native American TBI patients and propose possible explanations.

4. Inadequate Consideration of Cultural Factors:

The article acknowledges that Native American TBI patients may face issues such as cultural differences and distrust in the healthcare system, but it does not thoroughly investigate the impact of these cultural factors on recovery. It is recommended that the authors further explore the role of cultural factors in the recovery process and propose corresponding intervention strategies.

5. Directions for Future Research:

It is suggested that the authors propose future research directions, such as:

Collecting a larger sample size for more in-depth statistical analysis.

Exploring other factors that may affect the recovery of Native American TBI patients, such as comorbid conditions and geographical distance.

Reviewer #3: First of all, thanks a lot to the authors' research team for their efforts in this manuscript. I think this manuscript is innovative and extraordinarily clinically relevant. However, there are still some small problems with this manuscript that need further improvement:

1. In the Introduction section, the author describes the epidemiology, causes, and negative effects of traumatic brain injury in Native Americans and White Individuals. The “data differences” between the two in terms of prevalence, causes, and negative effects are also noted. However, this can still only be defined as a detailed introduction to the background, without clearly identifying the clinical significance of the study. I would suggest that the authors streamline some of the content in the introduction section and add a few points of necessity for the purpose of this study in the context of this background.

2. What the authors gained in the Results section was very fruitful, but they failed to have a full one-on-one discussion in the Discussion section.

3. Although the author makes many thought-provoking points in Global Outcome, it spans a wide range from the clinical to the societal level, so would it be possible to integrate them again in a separate paragraph.

4. In addition, perhaps the author should also pay attention to standardize some writing details, for example, “TBI” in the Introduction section should be expressed as Traumatic brain injury (TBI). Abbreviations in the Abstract should not be used directly in the text. Abbreviations that appear for the first time in the text should be labeled as such. (line 57)

There is also a problem with the formulation of this sentence： “Falls, collisions with an object, and motor vehicle accidents are leading causes of TBI in the U.S., while suicide, unintentional falls, and motor vehicle accidents are leading causes of TBI in the U.S.,” and the statement “TBI in the U.S. is the most common cause of TBI. unintentional falls, and motor vehicle accidents are the leading causes of TBI-related deaths in the U.S (1,4).” (lines 65-67)

There are other similar issues with writing conventions in the text, and it is hoped that the authors' reworking will make this manuscript a greater improvement in the presentation of its content.

Reviewer #4: I would like to thank the authors for submitting this manuscript. I believe the study has potential, especially given the lack of research on TBI and Native Americans. However, I think the research question needs to be better formulated, as it currently relies on hypotheses that are not well-supported scientifically. I recommend refining the hypotheses and research questions and supporting them with a more thorough literature review. Additionally, the sample size used in this study (75 Native Americans + 75 White individuals) seems too small to draw the conclusions presented in the manuscript, which is a significant limitation of the study. Lastly, in the introduction, I suggest rephrasing the paragraph discussing TBI and Native Americans, as the current wording in the manuscript may come across as insensitive to this group. I recommend rewording the paragraph in a more considerate manner, using as inclusive language as possible. Further comments can be found in the attached file.

6. PLOS authors have the option to publish the peer review history of their article (what does this mean?). If published, this will include your full peer review and any attached files.

Reviewer #1: No

Reviewer #2: **Yes: **bei li

Reviewer #3: No

Reviewer #4: No

---

## [Author Response · Author response to Decision Letter 1]

9 Oct 2024

Reviewer #1:

This paper was easy to read, used appropriate methods to assess the hypotheses, and addressed limitations as well as addressed future directions for research and clinical considerations for health care professionals. As such, this paper should assist clinicians and researchers in this area for future research and improvement in healthcare delivery for the relevant populations.

Thank you so much for your kind words and positive review of our manuscript. We are thrilled that you think it is a meaningful contribution to the literature!

Reviewer #2:

Comments from the editorial office: Upon internal evaluation of the reviews provided, we kindly request you to disregard the reviewer report provided by Reviewer 2. No amendments are required in response to Reviewer 2’s comments.

We have followed the editorial office’s guidance to disregard this reviewer report.

Reviewer #3:

First of all, thanks a lot to the authors' research team for their efforts in this manuscript. I think this manuscript is innovative and extraordinarily clinically relevant. However, there are still some small problems with this manuscript that need further improvement:

Thank you so much for your positive comments especially regarding the innovation and clinical relevance!

1. In the Introduction section, the author describes the epidemiology, causes, and negative effects of traumatic brain injury in Native Americans and White Individuals. The “data differences” between the two in terms of prevalence, causes, and negative effects are also noted. However, this can still only be defined as a detailed introduction to the background, without clearly identifying the clinical significance of the study. I would suggest that the authors streamline some of the content in the introduction section and add a few points of necessity for the purpose of this study in the context of this background.

Thank you for this comment. We have now added the following language to the Study Purpose section, highlighting key gaps in the literature and how our study helps fill these gaps: “Given the extensive differences in the causes of TBI, comorbidities, and outcomes between Native Americans and White individuals, the current study utilized a modified version of the MHDD (Figure 1) which has been previously used to examine disparities between Native Americans and White individuals with TBI (32) to investigate the degree to which disparities in global outcomes exist between the two groups over the 5 years post-injury. Further, only one other study to date has investigated how demographic and TBI injury characteristics differ between Native Americans and White individuals in a demographically matched sample (32). The current study highlighted demographic and injury-related characteristics that differed between the two groups then used these statistically significant differences to attempt to account for the disparity in global outcomes.”

2. What the authors gained in the Results section was very fruitful, but they failed to have a full one-on-one discussion in the Discussion section.

The primary results for the current study were: (1) Native Americans were less likely to be employed at the time of injury than White individuals, (2) Native Americans were less likely to have private insurance at the time of injury than White individuals, (3) Native Americans were more likely to report having used illicit/non-prescription drugs in the month prior to injury than White individuals, (4) Native Americans were more likely to report binge drinking in the month prior to injury than White individuals, (5) Native Americans reported worse global outcome scores both generally and over time, and (6) the difference in global outcome scores, both generally and over time, was accounted for by the addition of the sociodemographic characteristics that differed between the two groups.

The first paragraph of the Sociodemographic Differences section now discusses the results that Native Americans were less likely to be employed at the time of injury than White individuals. We have now added lines regarding private health insurance to greater emphasize these findings and offer a theory about how it might impact TBI rehabilitation.

The second paragraph under the Sociodemographic Differences section discusses substance use as a whole (i.e., both illicit non-prescription drug use and alcohol use). We have now reworked this paragraph to ensure this is more clearly stated. The differences in global outcome are now discussed in detail over the next two pages. We hope these changes help create a better one-on-one discussion of the results.

3. Although the author makes many thought-provoking points in Global Outcome, it spans a wide range from the clinical to the societal level, so would it be possible to integrate them again in a separate paragraph.

In the hopes of improving readability and integrating the results section into one, cohesive paragraph, the following paragraph was added to the end of the Discussion section: “The MHDD theorizes that health-related outcomes for individuals with disability are the result of interactions between a wide variety of factors. The current study reinforces this model by providing a picture of disparities in global outcome between Native Americans and White individuals with TBI that is complex and intersectional. Employment probabilities, type of insurance, illicit/non-prescription drugs, alcohol use, and systemic barriers may all play a role in predicting global outcome scores and disparities for Native Americans with TBI. Type of insurance appears to be particularly important as it was the only predictor to remain significant across all models.”

4. In addition, perhaps the author should also pay attention to standardize some writing details, for example, “TBI” in the Introduction section should be expressed as Traumatic brain injury (TBI). Abbreviations in the Abstract should not be used directly in the text. Abbreviations that appear for the first time in the text should be labeled as such. (line 57)

We have scanned the document and found two abbreviations used in the abstract that were not also expressed in the text of the manuscript. TBI has now been expressed as traumatic brain injury in the opening paragraph of the introduction, and TBIMS has been expressed as the Traumatic Brain Injury Model Systems in the first paragraph of the Procedure section.

There is also a problem with the formulation of this sentence: “Falls, collisions with an object, and motor vehicle accidents are leading causes of TBI in the U.S., while suicide, unintentional falls, and motor vehicle accidents are leading causes of TBI in the U.S.,” and the statement “TBI in the U.S. is the most common cause of TBI. unintentional falls, and motor vehicle accidents are the leading causes of TBI-related deaths in the U.S (1,4).”

We have now changed both of these to: “Falls, collisions with an object, and motor vehicle accidents are leading causes of TBI in the U.S., while suicide, unintentional falls, and motor vehicle accidents are the leading causes of TBI-related deaths in the U.S.”

There are other similar issues with writing conventions in the text, and it is hoped that the authors' reworking will make this manuscript a greater improvement in the presentation of its content.

We have conducted a thorough review of the document for other potential deviations from writing convention and corrected them wherever found.

Reviewer #4:

I would like to thank the authors for submitting this manuscript. I believe the study has potential, especially given the lack of research on TBI and Native Americans.

Thank you for reading our manuscript and for your comments recognizing the significant gap in the literature our research addresses.

However, I think the research question needs to be better formulated, as it currently relies on hypotheses that are not well-supported scientifically. I recommend refining the hypotheses and research questions and supporting them with a more thorough literature review.

We have changed the Hypotheses section to a newly title Study Rationale and Hypotheses section. We have added the following sentences to help integrate the information provided in the background to more cohesively justify our hypotheses. Additionally, while no information currently exists on global outcome trajectories, we have added four additional citations from the last 2 years justifying our approach through literature on other marginalized populations within the U.S. This section reads: “While no research to date has investigated global outcome for Native Americans, previously literature has shown that Native Americans face a plethora of barriers to recovery including: higher rates of depression and anxiety, lower healthcare utilization and access, and lower functional independence following TBI (28,32,42,43), indicating Native Americans are at a greater risk for negative complications following TBI when compared to White individuals. Further, while little research to date has investigated possible predictors of outcome following TBI for Native Americans, the literature for outcomes following TBI for other underserved populations (e.g., Hispanics) is growing, with evidence that sociodemographic factors like employment, alcohol use, marital status, violence as a cause of injury (47–50). Based on this literature, we hypothesize that…”

Additionally, the sample size used in this study (75 Native Americans + 75 White individuals) seems too small to draw the conclusions presented in the manuscript, which is a significant limitation of the study.

To address this concern, we have done several things. (1) We have added the following sentence to the Procedure and Study Approval section: “At present, no other database offers such nuanced data on Native Americans with TBI.” This sentence is intended to highlight the incredible uniqueness of the sample to study a construct that has never been investigated for Native Americans. (2) We have gone through the discussion section to attenuate the language surrounding the results, ensuring that the language does not imply causality. We have also extended the discussion section based on feedback from Reviewer 3 to include more detailed analysis of the results. (3) We have extended the limitation section to note (a) the uniqueness of the sample and (b) further expand the limitations on generalizability.

Lastly, in the introduction, I suggest rephrasing the paragraph discussing TBI and Native Americans, as the current wording in the manuscript may come across as insensitive to this group. I recommend rewording the paragraph in a more considerate manner, using as inclusive language as possible.

We have reviewed the section on TBI in Native Americans and have changed the language to reflect a more “ethnic” rather than “racial” approach as your line-items suggest. Additionally, we have added a few more citations to give the background literature greater context and assist with interpretation.

Further comments can be found in the attached file.

Line 38: Would "ethnic group" be a better term? I understand that in the U.S. there is the concept of racial medicine, but since this manuscript will be read internationally, I suggest using a more inclusive term like "ethnic group."

We have changed this to “ethnic group.”

Line 49: "Ethnic group."

We have changed this to ethnic group.

Line 59: Please include the acronym for the CDC.

We have no added “Centers for Disease Control and Prevention.”

Line 104: "Ethnic group."

We have changed this to “ethnic group.”

Line 107: "Ethnic minorities."

We have changed this to “ethnic minorities.”

Lines 116-117: When hypothesizing higher rates of TBI within the Native American population, only papers from 2004 are cited. Could you please include more recent references?

The references you have noted are specifically for higher rates of IPV related to TBI. In response to the reviewer’s question, we performed an additional very thorough search of the literature. Unfortunately, there are no more contemporaneous citations for this data. At the start of the “TBI in Native Americans” section, 6 citations are listed within the first two sentences that range in publication date from 2003-2023 in an attempt to showcase that the disproportionate rates of TBI have existed for at least the last 20 years or so.

Line 118: "Ethnic gender."

We have changed this to “ethnic gender.”

Line 119: "Ethnic group."

We have changed this to “ethnic group.”

Line 120: Could you please add more citations to support your argument?

We have now added an additional citation from the CDC that details violence as a cause of injury and an additional citation from 2016 that discusses IPV for Native Americans and how it may relate to increased risks of TBI.

Line 121: "Ethnic group." Additionally, to further support your argument, could you please reference other sources beyond (35-38)? Perhaps more recent studies published in high-impact journals.

We have changed this to “ethnic groups.” We have added a citation from 2020 discussing the possible link between alcohol use disorder and TBI. Unfortunately, substance use and TBI are grossly under-researched in Native Americans. The citations used in the manuscript are the most recent on this topic and are all from high impact journals within the last ten years.

Line 124: A citation is missing. Since we are still in the introduction, personal opinions or hypotheses should not be presented at this point in the manuscript.

We have now added a citation that discusses the links between alcohol use disorder, impaired driving, and TBI.

Line 150: "Ethnic health."

We have changed this to “ethnic health.”

Line 158: Please avoid referring to figures present in the manuscript within the introduction section.

We have now moved Figure 1 to the Study Measures - Demographic and Injury-Related Variables of the Method.

Line 159: Why is this citation included? You are discussing a model that has been utilized in the current study.

We have added the phrase “which has been previously” to the sentence to highlight that this model has a history of being used to investigate health disparities for Native Americans with TBI.

Line 163-170: It might be helpful to present the hypotheses and the research objective in a less schematic and more fluid, linear manner.

We have integrated the hypotheses into a paragraph and more fluid context.

Line 167: I would suggest removing Hypothesis 3 since it is not tested in this study.

We apologize for the confusion, but Hypothesis 3 is tested in the current study via the hierarchical linear multiple regressions run during the analyses. Primary Set 2 incorporates the significantly different sociodemographic variables between the two groups in an attempt to account for the variance in global outcome scores. The results demonstrate that Hypothesis 3 was rejected; that is, the difference in global outcomes scores did not persist after accounting for the significantly different sociodemographic variables between the two groups.

Lines 199-202: This section may be better suited for the Results rather than the Methods. The Methods should specify only the statistical tests used, without previewing the results.

We appreciate this commentary; however, this section deals directly with how we reached a sample size of 75 Native Americans and 75 White individuals with TBI. We believe this section fits most fluidly within the participants section as it is not part of the study results but how we formed our sample. If the reviewer or editor feels strongly that this should be moved, we are happy to do so.

Line 200: Before performing the ANOVA test, were the data checked for normal distribution?

We did not check the year of injury ANOVA for normal distribution, but upon a second reading, we actually do not think this exploratory analysis is important. We have now deleted it to streamline the manuscript better, as requested by the reviewers.

Line 212: "Ethnicity."

We have changed this to “ethnicity.”

Lines 213-216: I believe this line may not be necessary for the manuscript.

We have deleted t

---

## [Decision Letter · Decision Letter 1]

4 Dec 2024

PONE-D-24-26489R1Disparities Between Native Americans and White Individuals in Global Outcome Trajectories over the 5 Years after Traumatic Brain Injury: A Model Systems StudyPLOS ONE

Dear Dr. Perrin,

Thank you for submitting your manuscript to PLOS ONE. After careful consideration, we feel that it has merit but does not fully meet PLOS ONE’s publication criteria as it currently stands. Therefore, we invite you to submit a revised version of the manuscript that addresses the points raised during the review process.

Two reviewers (#1 and #3) in the first round of the review who requested minor revisions were satisfied with the revised version. The third reviewer (#4) in the first first round of the review who requested major revisions was not able to review this revised version. Therefore, one additional reviewer (#5) was invited and this reviewer pointed out more major issues in this study. Please revise your manuscript according to the comments from reviewer #5. Below is the comments from reviewer #5:

1. The revised introduction adds context to the study’s significance but fails to connect research gaps to actionable clinical or policy objectives. It focuses too heavily on statistical analysis while neglecting systemic barriers like access to culturally informed care or tribal healthcare limitations. Without addressing these issues, the introduction weakens the case for why these disparities matter for healthcare interventions and policy-making.

2. Discussions of employment, insurance, and substance use findings were added, but they lack depth and fail to analyze systemic factors like geographic barriers or socioeconomic disadvantages. Private insurance is identified as a predictor, but its role within broader inequities is unexplored. The absence of unmeasured variables, such as healthcare trust or systemic racism, limits the analysis of disparities in long-term care for Native Americans.

3. The consolidation of the global outcomes discussion improves readability but does little to deepen the analysis. The authors fail to explain why disparities worsen over time or suggest interventions to address them. Specific components of the Glasgow Outcome Scale-Extended (e.g., family and social roles) that may disproportionately affect Native Americans remain unexamined, leaving a critical gap in understanding disparities.

4. Revising the hypotheses and adding citations improves their context but fails to address concerns about causality. Hypothesis 3 remains untested, and it is unclear whether the added citations are specific to Native Americans or apply broadly to underserved populations. Explicitly acknowledging limitations in inferring causality would improve the framing of the hypotheses.

5. The authors justify the small sample size as reflective of the rarity of Native American TBI cases, but this does not address its impact on generalizability and statistical power. Power analyses are not discussed, and sensitivity analyses could help demonstrate the robustness of the findings despite this limitation. These omissions weaken the study’s credibility.

We look forward to receiving your revised manuscript.

Kind regards,

Leming Zhou

Academic Editor

PLOS ONE

Reviewers' comments:

Reviewer's Responses to Questions

**Comments to the Author**

1. If the authors have adequately addressed your comments raised in a previous round of review and you feel that this manuscript is now acceptable for publication, you may indicate that here to bypass the “Comments to the Author” section, enter your conflict of interest statement in the “Confidential to Editor” section, and submit your "Accept" recommendation.

Reviewer #1: All comments have been addressed

Reviewer #3: All comments have been addressed

Reviewer #5: (No Response)

2. Is the manuscript technically sound, and do the data support the conclusions?

Reviewer #1: Yes

Reviewer #3: Yes

Reviewer #5: Yes

3. Has the statistical analysis been performed appropriately and rigorously? 

Reviewer #1: Yes

Reviewer #3: Yes

Reviewer #5: No

4. Have the authors made all data underlying the findings in their manuscript fully available?

Reviewer #1: Yes

Reviewer #3: Yes

Reviewer #5: No

5. Is the manuscript presented in an intelligible fashion and written in standard English?

Reviewer #1: Yes

Reviewer #3: Yes

Reviewer #5: Yes

6. Review Comments to the Author

Reviewer #1: You have completed all revisions satisfactorily which in turn has made the paper even better. Excellent work.

Reviewer #3: (No Response)

Reviewer #5: 1. The revised introduction adds context to the study’s significance but fails to connect research gaps to actionable clinical or policy objectives. It focuses too heavily on statistical analysis while neglecting systemic barriers like access to culturally informed care or tribal healthcare limitations. Without addressing these issues, the introduction weakens the case for why these disparities matter for healthcare interventions and policy-making.

2. Discussions of employment, insurance, and substance use findings were added, but they lack depth and fail to analyze systemic factors like geographic barriers or socioeconomic disadvantages. Private insurance is identified as a predictor, but its role within broader inequities is unexplored. The absence of unmeasured variables, such as healthcare trust or systemic racism, limits the analysis of disparities in long-term care for Native Americans.

3. The consolidation of the global outcomes discussion improves readability but does little to deepen the analysis. The authors fail to explain why disparities worsen over time or suggest interventions to address them. Specific components of the Glasgow Outcome Scale-Extended (e.g., family and social roles) that may disproportionately affect Native Americans remain unexamined, leaving a critical gap in understanding disparities.

4. Revising the hypotheses and adding citations improves their context but fails to address concerns about causality. Hypothesis 3 remains untested, and it is unclear whether the added citations are specific to Native Americans or apply broadly to underserved populations. Explicitly acknowledging limitations in inferring causality would improve the framing of the hypotheses.

5. The authors justify the small sample size as reflective of the rarity of Native American TBI cases, but this does not address its impact on generalizability and statistical power. Power analyses are not discussed, and sensitivity analyses could help demonstrate the robustness of the findings despite this limitation. These omissions weaken the study’s credibility.

7. PLOS authors have the option to publish the peer review history of their article (what does this mean?). If published, this will include your full peer review and any attached files.

Reviewer #1: No

Reviewer #3: No

Reviewer #5: No

---

## [Author Response · Author response to Decision Letter 2]

6 Feb 2025

Comments to the Author

1. If the authors have adequately addressed your comments raised in a previous round of review and you feel that this manuscript is now acceptable for publication, you may indicate that here to bypass the “Comments to the Author” section, enter your conflict of interest statement in the “Confidential to Editor” section, and submit your "Accept" recommendation.

Reviewer #1: All comments have been addressed

Reviewer #3: All comments have been addressed

Reviewer #5: (No Response)

2. Is the manuscript technically sound, and do the data support the conclusions?

Reviewer #1: Yes

Reviewer #3: Yes

Reviewer #5: Yes

3. Has the statistical analysis been performed appropriately and rigorously?

Reviewer #1: Yes

Reviewer #3: Yes

Reviewer #5: No

4. Have the authors made all data underlying the findings in their manuscript fully available?

Reviewer #1: Yes

Reviewer #3: Yes

Reviewer #5: No

We have now stated in the Procedure and Study Approval section that “The data are publicly available through an external data request to the TBIMS National Data and Statistical Center (https://www.tbindsc.org/researchers.aspx).”

5. Is the manuscript presented in an intelligible fashion and written in standard English?

Reviewer #1: Yes

Reviewer #3: Yes

Reviewer #5: Yes

6. Review Comments to the Author

Reviewer #1: You have completed all revisions satisfactorily which in turn has made the paper even better. Excellent work.

Reviewer #3: (No Response)

Reviewer #5:

1. The revised introduction adds context to the study’s significance but fails to connect research gaps to actionable clinical or policy objectives. It focuses too heavily on statistical analysis while neglecting systemic barriers like access to culturally informed care or tribal healthcare limitations. Without addressing these issues, the introduction weakens the case for why these disparities matter for healthcare interventions and policy-making.

We have now added information on a study that investigates disparities between Native Americans and White individuals for community participation following TBI. We have also added the following paragraph to the Introduction section to draw attention to general disparities/marginalization, healthcare specific disparities, and issues around cultural dissonance and provider mistrust faced by Native Americans:

“In addition to inequal outcomes following TBI, Native Americans are recognized as a severely underserved and marginalized population, facing disparities in diseases, healthcare utilization, poverty rates, and lower education and employment (46). Native Americans are significantly more likely to face premature mortality due in large part to disparities in chronic disease and lack of appropriate specialized healthcare providers in tribal lands or through the Indian Health Service (47). The lack of reliable and adequate healthcare services, heightened rates of high mortality-related injuries and diseases (e.g., TBI, diabetes), and suspicion of western medical practices and provider distrust, combine to create a bleak picture for the state of Native American healthcare (48,49).”

2. Discussions of employment, insurance, and substance use findings were added, but they lack depth and fail to analyze systemic factors like geographic barriers or socioeconomic disadvantages. Private insurance is identified as a predictor, but its role within broader inequities is unexplored. The absence of unmeasured variables, such as healthcare trust or systemic racism, limits the analysis of disparities in long-term care for Native Americans.

The first paragraph to the Sociodemographic Differences section was significantly updated and extended with several new citations and additional information as follows:

“In line with previous research suggesting Native Americans experience lower employment rates than the general U.S. population and within the TBI population (32,33,45), the current study found that Native Americans were significantly less likely to be employed prior to experiencing a TBI than their White counterparts. While information on employment and insurance rates for Native Americans is lacking, particularly for Native Americans with TBI, a separate TBIMS study utilizing a sample that partially overlaps with the present study also suggested Native Americans were less likely to be employed prior to injury and less likely to have private insurance than White individuals (32). These may be key contributing factors for why Native Americans are more likely to experience barriers to healthcare access and less likely to engage with the healthcare system as private insurance often affords greater and more varied access to care, and employment is often a primary means of acquiring health insurance (28,62). Private health insurance also typically affords greater and more varied access to healthcare than non-private alternatives. It is possible that Native Americans who lack private healthcare experience worse care or have a more difficult time accessing appropriate care, thus negatively impacting their recovery post-TBI. Given the higher rates of poverty within the Native American population coupled with their documented concerns about the high cost of healthcare, this may make Native Americans less likely to seek healthcare services even if they are available (48,63). Issues such as cultural dissonance, provider distrust, and suspicion of Western medical practices may further impede Native Americans’ decision to seek treatment (47,48). These concerns, coupled with higher rates of TBI and chronic comorbid conditions (e.g., diabetes) create a scenario in which Native Americans may need greater and more specialized healthcare but be less likely to engage (32,33,47,63).”

Further, we added additional information to the limitations section to highlight that the current study was unable to explore issues surrounding systemic barriers and highlighted specific variables that may be of interest for future studies: “Research indicates that other variables not assessed in the current study like systemic barriers (e.g., distance to the nearest hospital), quality of care provided, the number and severity of comorbid conditions, cultural dissonance between patient and provider may be important factors contributing to differential outcomes following TBI (32,33,45,47,48). Particularly given the limited research on racial experiences of Native Americans with healthcare systems and providers (70), future research may wish to investigate how racial bias may impact healthcare access and quality for Native Americans with TBI.”

This added section ties in directly with a section added to the Global Outcome section that directly addresses issues of systemic barriers/racism: “Finally, while not assessed in the present study, a plethora of research has indicated racism as a key factor contributing to healthcare inequalities (66–68). Especially within the U.S., some research has suggested that racism is embedded within institutions, legal systems, and social structures and continues to negatively impact racial/ethnic minority communities who are unfairly targeted or marginalized by the U.S. healthcare system (69). The limited literature on Native Americans with TBI indicates they experience high levels of racial discrimination (70) and are more likely to receive invasive surgery whereas White individuals are less likely and instead to be recommended for less invasive treatments like occupational or physical therapy, despite comparable TBI pathology (38). This, coupled with disparities in various outcomes (e.g., cognitive and motor function, depression) can paint a bleak picture for the state of care received by Native Americans and may be contributing to lower global outcome scores (32,43).

3. The consolidation of the global outcomes discussion improves readability but does little to deepen the analysis. The authors fail to explain why disparities worsen over time or suggest interventions to address them. Specific components of the Glasgow Outcome Scale-Extended (e.g., family and social roles) that may disproportionately affect Native Americans remain unexamined, leaving a critical gap in understanding disparities.

The following paragraphs have been expanded with additional information and citation in the hopes of filling the gaps mentioned above: “Type of insurance was also identified in the current study as a key source of disparities in global outcome scores between Native Americans and White individuals with TBI. This makes sense as private insurance can provide greater access to care and more varied resources. The lack of resource may result in less specialized care or force Native Americans to use rural healthcare centers or Indian Health Service facilities that may be ill-equipped to provide the type and quality of care necessary for more complex conditions like TBI (47). Given that Native Americans frequently experience barriers to healthcare services (e.g., cultural mistrust, racism) and are less likely to utilize healthcare services and resources and to receive or be recommended for more advanced services post-rehabilitation (28,38), it is possible that type of insurance may limit access for Native Americans and thereby reduce global outcome scores. While not assessed in the present study, it is possible the lack of private insurance and reduced healthcare service utilization may lead to the development of additional comorbid conditions (46).

Finally, while not assessed in the present study, a plethora of research has indicated racism as a key factor contributing to healthcare inequalities (66–68). Especially within the U.S., some research has suggested that racism is embedded within institutions, legal systems, and social structures and continues to negatively impact racial/ethnic minority communities who are unfairly targeted or marginalized by the U.S. healthcare system (69). The limited literature on Native Americans with TBI indicates they experience high levels of racial discrimination (70) and are more likely to receive invasive surgery whereas White individuals are less likely and instead to be recommended for less invasive treatments like occupational or physical therapy, despite comparable TBI pathology (38). This, coupled with disparities in various outcomes (e.g., cognitive and motor function, depression) can paint a bleak picture for the state of care received by Native Americans and may be contributing to lower global outcome scores (32,43).”

The following two sentences were added to the second paragraph of the Limitations and Future Directions section to, as the reviewer suggested, note the limitation of using a total GOS-E score rather than examine item-level differences: “The use of the total score for the GOS-E reduces some of the nuance that may be obtained by exploring item-level differences. Future studies may wish to use a larger sample to analyze differences in specific components of the GOS-E.”

4. Revising the hypotheses and adding citations improves their context but fails to address concerns about causality. Hypothesis 3 remains untested, and it is unclear whether the added citations are specific to Native Americans or apply broadly to underserved populations. Explicitly acknowledging limitations in inferring causality would improve the framing of the hypotheses.

We apologize for our lack of clarity. The third hypothesis is tested by “Primary Set 2” in which we run a second set of HLMs that include the statistically different covariates. To help with clarity on which hypotheses are tested by the various analysis steps, we have added notation for “hypothesis 1,” “hypothesis 2,” and “hypothesis 3” throughout the results section. We have also explicitly stated in the Primary Set 2 section “thus, for the study’s third hypothesis, we failed to reject the null.”

Based on feedback from previous reviewers we expanded the section in Limitations and Future Directions regarding the inability to infer causality. That section now reads as follows: “While the current study identified employment at the time of injury, substance use, and type of insurance as important factors accounting for global outcome disparities between Native Americans and White individuals, the exact mechanism for these relations was not uncovered in the current study. Despite the large number of sociodemographic and injury-related characteristics investigated, it is impossible to prove a causal relationship between these covariates and global outcome disparities. That is, the significantly different variables between the two groups could be markers for other unmeasured sources of variance that directly contribute to disparities in global outcome following TBI for Native Americans.”

5. The authors justify the small sample size as reflective of the rarity of Native American TBI cases, but this does not address its impact on generalizability and statistical power. Power analyses are not discussed, and sensitivity analyses could help demonstrate the robustness of the findings despite this limitation. These omissions weaken the study’s credibility.

To address the reviewer’s concerns regarding statistical power, we have added the following new information and analyses to the Results section:

“Effect Sizes and Post-hoc Power Analysis. Cohen’s d effect sizes were calculated for the differences between Native American and White Individuals’ GOS-E scores at years 1, 2, and 5 post-injury (Table 4). Year one was below the threshold of a small-sized effect; year two showed a small-sized effect, and year five showed a medium sized-effect. A post-hoc power analysis conducted a priori in G*Power 3.1.9.7 targeting a medium effect size with α = .05 and power at 0.80, revealed the need for a total sample size at N = 102, well below the 150 sample for the current study.

Table 4

Effect Sizes for GOSE at 1-, 2-, and 5-Years Post-injury by Race/Ethnicity

Variable Native Americans White Individuals

 Cohen's d M (SD) M (SD)

One Year GOSE 0.17 5.42 (1.84) 5.72 (1.68)

Two Year GOSE 0.20 5.42 (1.80) 5.78 (1.77)

Five Year GOSE 0.58 4.95 (1.98) 5.98 (1.65)

Additionally, we have added the following details to the Limitations and Future Directions section in which we explicitly mention the difficulties surrounding generalizability as well as cite literature on the robustness of HLM and how the sample size of 150 individuals is more than adequate for the present analyses: “While the sample size of 150 Native American and White individuals is adequate for the present analyses, large enough to detect a medium effect size, (71,72) and unique w

---

## [Decision Letter · Decision Letter 2]

4 Mar 2025

Disparities Between Native Americans and White Individuals in Global Outcome Trajectories over the 5 Years after Traumatic Brain Injury: A Model Systems Study

PONE-D-24-26489R2

Dear Dr. Perrin,

We’re pleased to inform you that your manuscript has been judged scientifically suitable for publication and will be formally accepted for publication once it meets all outstanding technical requirements.

Kind regards,

Leming Zhou

Academic Editor

PLOS ONE

Additional Editor Comments (optional):

Reviewers' comments:

Reviewer's Responses to Questions

**Comments to the Author**

1. If the authors have adequately addressed your comments raised in a previous round of review and you feel that this manuscript is now acceptable for publication, you may indicate that here to bypass the “Comments to the Author” section, enter your conflict of interest statement in the “Confidential to Editor” section, and submit your "Accept" recommendation.

Reviewer #1: All comments have been addressed

Reviewer #3: All comments have been addressed

Reviewer #5: All comments have been addressed

2. Is the manuscript technically sound, and do the data support the conclusions?

Reviewer #1: Yes

Reviewer #3: Yes

Reviewer #5: Yes

3. Has the statistical analysis been performed appropriately and rigorously? 

Reviewer #1: Yes

Reviewer #3: Yes

Reviewer #5: Yes

4. Have the authors made all data underlying the findings in their manuscript fully available?

Reviewer #1: Yes

Reviewer #3: Yes

Reviewer #5: Yes

5. Is the manuscript presented in an intelligible fashion and written in standard English?

Reviewer #1: Yes

Reviewer #3: Yes

Reviewer #5: Yes

6. Review Comments to the Author

Reviewer #1: I have no additional comments from the previous reviews. the authors have revised the paper satisfactorily.

Reviewer #3: (No Response)

Reviewer #5: The authors have addressed all my concerns. I have no additional comments.

7. PLOS authors have the option to publish the peer review history of their article (what does this mean?). If published, this will include your full peer review and any attached files.

Reviewer #1: No

Reviewer #3: **Yes: **Gong Cheng

Reviewer #5: No

---

## [Editor Report · Acceptance letter]

PONE-D-24-26489R2

PLOS ONE

Dear Dr. Perrin,

I'm pleased to inform you that your manuscript has been deemed suitable for publication in PLOS ONE. Congratulations! Your manuscript is now being handed over to our production team.

Kind regards,

on behalf of

Dr. Leming Zhou

Academic Editor

PLOS ONE